

**Soil bacterial community triggered by organic matter inputs**
**supports a high-yielding pear production**
Li Wang[1], Xiaomei Ye[1✉], Hangwei Hu[2], Jing Du[1], Yonglan Xi[1], Zongzhuan Shen[3✉], Jing Lin[4], Deli
Chen[2]
**Affiliations**
[1] Institute of Animal Science, Jiangsu Academy of Agricultural Sciences, Key Laboratory of Crop and
Livestock Integrated Farming, Ministry of Agriculture and Rural Affairs, Nanjing 210014, China
[2] Faculty of Veterinary and Agricultural Sciences, University of Melbourne, Melbourne, Victoria 3010,
Australia
[3] Jiangsu Provincial Key Lab of Solid Organic Waste Utilization, Jiangsu Collaborative Innovation
Center of Solid Organic Wastes, Educational Ministry Engineering Center of Resource-saving fertilizers,
The Key Laboratory of Plant Immunity, Nanjing Agricultural University, Nanjing, 210095, China
[4] Institute of Pomology, Jiangsu Academy of Agricultural Sciences, Nanjing 210014, China
✉ Corresponding author: Xiaomei Ye, Institute of Animal Science, Jiangsu Academy of Agricultural
Sciences, Key Laboratory of Crop and Livestock Integrated Farming, Ministry of Agriculture and Rural
Affairs, Nanjing 210014, China. Zognzhuan Shen, College of Resources and Environmental Sciences,
Nanjing Agricultural University, 210095, Nanjing, China. E-mail: shenzongz@njau.edu.cn. Tel:

21 (86)02584395521.

**Abstract.** The roles of microorganisms in enhancing crop production have been demonstrated for a range
of cropping systems. Most studies to date, however, have been confined to a limited number of locations,
making it difficult to identify general soil biotic and abiotic characteristics underpinning the yield-
promotion across various locations. This knowledge gap limits our capacity to harness soil microbiome
to improve crop production. Here we used high-throughput amplicon sequencing to investigate the
common features of bacterial community composition, ecological networks and physicochemical
properties in six yield-invigorating and adjacent yield-debilitating orchards. We found that yield-





invigorating soils exhibited higher contents of organic matter than yield-debilitating soils and harboured
unique bacterial communities. Greater alpha diversity and higher relative abundances of Planctomycetes
and Chloroflexi were observed in yield-debilitating soils. Co-occurrence network analysis revealed that
yield-invigorating soils displayed a greater number of meta-modules and a higher proportion of negative
links to positive links. Chloroflexi was recognized as a keystone taxon in manipulating the interaction of
bacterial communities in yield-invigorating soils. Structural equation modelling showed that soil organic
matter, beta diversity of bacterial community, and network connector (Chloroflexi) were key factors
supporting high-yield pear production. Altogether, we provide evidence that yield-invigorating soils
across a range of locations appear to share common features, including accumulation of soil organic
matter, higher microbial diversity, enrichment of key taxa like Chloroflexi, and maintaining a competitive
network. These findings have implications for science-based guidance for sustainable food production.

**Keywords:** Soil organic matter, Microbial diversity, Random forest prediction, Co-occurrence network,
Keystone taxa

**1    Introduction**
Soils are essential to human wellbeing due to their great contributions to the production of food, fiber,
feed, and medicine (Raaijmakers and Mazzola, 2016). Soil organisms play critical roles in maintaining
these ecosystem services, such as driving nutrient cycling, maintaining soil fertility, improving plant
productivity and suppressing plant diseases (Bender et al., 2016; Barrios, 2007). Microorganisms
participate in nearly all soil biological processes, and the microbial abundance, community composition
and activity primarily determine the sustainable productivity of agricultural lands (Philippot et al., 2013).
Given that bacteria are the most diverse and abundant group of microorganisms in soil, bacterial
communities and their functions can be pivotal indicators for crop production in agroecosystems (van
der Heijden et al., 2008).

In general, an increase in microbial diversity is linked to a high-yielding crop production mainly

through improving the host resilience to physical or chemical disturbances, modifying plant competition,
and facilitating plant access to nutrients (Chaer et al., 2009, Kennedy and Smith, 1995). Since individual
organisms do not live in isolation but rather form a complex system of inter-species interactions in soil,
interactions among community members were found to be related to crop production in the monoculture



system (Lu et al., 2013). Enrichments of key functional microbes in soil were deemed to serve specific
soil system functions, such as suppressing soil-borne pathogens and maintaining sustainable crop
production (Banerjee et al., 2018). However, the relative contributions of microbial diversity, interactions
among community members, or enrichment of key taxa to crop production remain largely unknown.
Therefore, it is highly desirable to identify pivotal indicators of bacterial community composition in
response to high-yielding crop production.
Changes in composition of soil bacterial communities across space are often strongly correlated
with soil pH (Fierer and Jackson, 2006). Soil pH has been recognized as a key driver in determining the
assembly of bacterial community in arable soils by field or microcosm experiments (Rousk et al., 2010).
However, recent studies have demonstrated that compositions of soil bacterial communities were driven
by a myriad of soil abiotic traits, such as organic matter contents, nutrient contents and forms (Tian et
al., 2018; Wang et al., 2018). For example, soil bacterial community composition, which determines the
ability of soil to suppress soil-borne pathogens, was found to be strongly correlated with soil organic
matter (Shen et al., 2018). An imbalanced ratio of soil nutrients, *i.e.*, ratio of nitrogen to phosphorus or
potassium could be a driving force altering the bacterial community composition in long-term fertilized
soils (Eo and Park, 2016). Therefore, key soil chemical properties identified in controlling the
distribution and abundance of bacterial community is largely depending on soil sampling scale or
treatments. As a consequence, a better understanding of the relationship between soil edaphic properties
and bacterial community composition is critical to develop targeted manipulation options to increase soil
service provisions.
Pear (*Pyrus*) is the third most important temperate fruit species second only to grape and apple,
belonging to the subfamily *Pomoideae* in the family *Rosaceae*. As a popular fruit in the world market,
pear has been cultivated globally, and China is the biggest pear producer (FAOSTAT, 2019). 'Sucui No.
1' pear, an early-maturing cultivated variety bred by the Jiangsu Academy of Agricultural Sciences,
China, has displayed distinct advantages over other cultivates in Eastern and Central China, because this
variety is easy to produce, adaptable to the environment, and has good quality and high economic benefits
(Lin et al., 2013). With the increasing demand in China, sustainable production of high-quality pear is
becoming increasingly important. Manipulation of soil microbiomes has shown to be an effective way
to increase soil productivity (Chaparro et al., 2012). Considering that large-scale surveys could exhibit
the diversity of soil microbial communities exceeds what is found in host-associated communities (Zorz



et al., 2019), it is necessary to explore the general microbial characteristics of multiple yield-invigorating
soils and identify key environmental drivers in assembling bacterial community.

In this study, six yield-invigorating and adjacent yield-debilitating pear orchards, which were

identified through field surveys, were selected. We hypothesized that yield-invigorating pear orchard
soils harbor unique bacterial communities which are manipulated by key soil abiotic factors. To address
this, soil bacterial communities and edaphic properties of six yield-invigorating and adjacent yield-
debilitating pear orchards were compared to (1) decipher the differences of taxonomic diversity, and
composition of the bacterial community, and (2) determine the contributions of environmental variables
to the changes in the structure of bacterial communities.
**2    Methods**
2.1   Study sites and experimental design
From July - August 2019, a field production survey of orchards cultivated with 'Suci No. 1' pear was
performed after pear fruits harvest to compare the differences of soil nutrients and microbiota between
yield-invigorating (YI) with yield-debilitating (YD) orchards. The locations, planting density, cropping
years, soil type and total yield were recorded. To minimize the effects of microclimate at each site, only
pair-located pear orchards with invigorating and debilitating yield and at similar growth stage were
selected for this research. In total, six pair-located yield-invigorating and -debilitating pear orchards
distributed in four cities of Jiangsu province, China, were selected in the main pear production areas (Fig.
1A, Table S1).

Paired yield-invigorating and yield-debilitating orchards from Fengxian (FX), Suining (SN) and

Tongshan (TS) were maintained in the Xuzhou city under the warm temperate sub-humid monsoon
climate. This site has a mean annual temperature (MAT) of 14.5 °C and mean annual precipitation (MAP)
of 847 mm. Orchards from Taixing (TX) were located in the Taizhou city under the humid southern
subtropical climate with a MAT of 15.3 °C and MAP of 1055 mm. Orchards from Gaochun (GC) were
located in the Nanjing city under the humid subtropical monsoon climate with a MAT of 15.4 °C and
MAP of 1106 mm. Orchards from Zhangjiagang (ZJ) were located in the Suzhou city under the humid
subtropical monsoon climate with a MAT of 15.7 °C and MAP of 1094 mm. For paired yield-invigorating
and-debilitating orchards, the irrigation and pesticide management practices were similar according to
farm records. However, yield-invigorating orchard was usually amended with more organic fertilizer
under integrated nutrients management whereas the co-located yield-debilitating orchard received more



chemical fertilizer under intensive management. The yield per tree was obtained by dividing the total
yield per hectare by plant density. Detailed information about each orchard is shown in Table S1.
2.2  Soil sample collection and chemical properties determination
Along with the field survey, soil sampling campaigns were performed from July - August 2019 after pear
fruits harvest. For each yield-invigorating or -debilitating orchard, four subplots with three pear trees in
each subplot were randomly selected for soil sampling. Subsequently three soil cores (0-20 cm) under
the trunk base for each tree were collected using a 25 mm soil auger. In total, nine soil cores for each
subplot were pooled as a composite sample and finally four composite soil samples for each orchard
were collected and promptly transported on ice to the laboratory. After sifting through a 2 mm sieve and
thoroughly mixing, one portion of each soil sample was air-dried for chemical property analyses while
the remainder was stored at -70 °C for DNA extraction.
Soil chemical properties including soil pH, content of organic matter (OM), total nitrogen (TN),
available phosphorus (AP), available potassium (AK), alkali-hydrolyzale nitrogen (N), exchangeable
calcium (Ca), effective magnesium (Mg), effective iron (Fe), effective manganese (Mn), effective copper
(Cu) and effective zinc (Zn), were measured according to methods described by Shen et al. (2018) and
Huang et al. (2019). Briefly, soil pH was determined using a glass electrode meter in a suspension with
a 1:5 soil/water ratio (w/v). Soil OM was determined using the potassium dichromate external heating
method. TN was determined using a dry combustion method on an Element Analyzer (Vario EL,
Germany). AP and AK were determined using the molybdenum blue method after soil was extracted with
sodium bicarbonate and flame photometry after soil was extracted with ammonium acetate, respectively.
Soil alkaline hydrolysable nitrogen (N) was measured by the alkaline hydrolysable diffusion method.
Contents of soil Ca, Mg, Fe, Mn, Cu and Zn were determined by the atomic absorption spectroscopy
method using ICE 3300 AAS Atomic Absorption Spectrometer (ThermoScientific, USA) after acid
hydrolysis.
2.3  Soil DNA extraction and bacterial abundance quantification
Genomic DNA from 0.25 g soil for each sample was extracted by using the DNeasy® PowerSoil® Kit
(QIAGEN GmbH, Germany) according to the manufacturer's instructions. The abundances of soil
bacteria were determined with Eub338F/Eub518R primer using a 7500 Real Time PCR System (Applied
Biosystems, USA). Standard curves were generated by using 10-fold serial dilutions of a plasmid
containing a full-length copy of the 16S rRNA gene from *Escherichia coli*. Quantitative PCR analysis



was performed in 96-well plates with a 20-µl mixture for each reaction using SYBR®Premix Ex Taq™
(TaKaRa, Japan). Thermal cycling was conducted according to a standard procedure with three replicates,
and the results were expressed as log copy numbers g$^{-1}$ dry soil.
2.4  Sequencing library construction and sequencing
The gene-specific primers 515F/806R with 12 bp barcode were used to amplify the V4 region of bacterial
16S rRNA gene on the BioRad S1000 (Bio-Rad Laboratory, CA) roughly according to the protocols
described by Caporaso et al. (2011). All constructed libraries were sequenced using the Illumina
NovaSeq 6000 at the Guangdong Magigene Biotechnology Co., Ltd. (Guangzhou, China).
2.5  Sequence processing
Quality filtering of the paired-end raw reads was performed to obtain the high-quality clean reads
according to the Trimmomatic (V0.33) quality control process. Sequences were assigned to each sample
based on their unique barcode, after which the barcodes and primers were removed. Paired-end clean
reads were merged using FLASH (V1.2.11). Raw tags were processed to generate the final ASV
(Amplicon Sequence Variant) table file at 97% pairwise identity according to the QIIME2 pipeline
(Bolyen et al., 2019). The nonbacterial and mitochondrial ASVs and extremely low frequency ASVs
(relative abundance < 0.01%) were removed. A representative sequence for each ASV was selected and
classified using the RDP classifier (Wang et al., 2007) against the RDP Bacterial 16S database.
2.6  Statistical analyses
Statistical analyses were performed using the software SPSS 20.0 and R. Non-normal data were square-
root or log transformed when necessary. The significance of soil properties or microbial taxa in yield-
invigorating or-debilitating orchards was determined based on paired Wilcoxon rank sum test, and
adjusted $P$ values (< 0.05) were obtained by the FDR method. Mantel tests were used to identify the
correlations between microbial community composition and pear yield, and soil chemical properties
using the 'vegan' package in R. The linear regression analyses relating yield to selected microbial taxa
or soil chemical properties were conducted using the 'basicTrendline' package in R.
Principal Coordinate Analysis (PCoA) based on the Bray-Curtis distance was performed in
MOTHUR (V1.38.1) (Schloss et al., 2009) and visualized by the 'ggplot2' package in R to explore the
differences in microbial community composition. Permutational multivariate analysis of variance
(PERMANOVA) was performed to evaluate the significant differences of microbial community
composition according to sample locations and orchard yield using the 'vegan' package in R. Microbial





alpha diversity indexes (Chao, Shannon) were calculated based on randomly resampled ASV abundance
matrices at the same depth (23,800 sequences) in MOTHUR. A Venn diagram was generated based on
the final ASVs to compare microbial community composition between yield-invigorating and -
debilitating orchard soils. The affiliations of unique and shared ASVs in yield-invigorating and -
debilitating soils were compared to evaluate the differences of the bacterial community composition and
plotted using the 'pheatmap' package in R. Fold changes (log2 transformed) of shared ASVs across
yield-invigorating and -debilitating soils were calculated. The ASVs with fold change ratios > 2 and
unique ASVs in yield-invigorating soils were recognized as potential responders to yield promotion. In
addition, to better understand the bacterial community composition, relative abundances at the genus
level were compared by STAMP software v2.1.3 (Parks et al., 2014).
Potential ecological interactions among bacteria were determined by modeling the microbial
community using Molecular Ecological Network Analysis (http://ieg2.ou.edu/MENA) based on pear
yield. After removal of ASVs whose abundances were lower than 0.01%, the ASVs table appeared in at
least half soil samples were merged for phylogenetic molecular ecological network (pMEN) construction
(Deng et al., 2012). The microbial network was constructed using random matrix theory-based at 0.94
similarity threshold and visualized using Cytoscape 2.8.3 software (V3.5.1, http://cytoscape.org/).
Module clustering and composition in yield-invigorating and -debilitating networks were compared and
plotted in R using the 'pheatmap' and 'ggplot2' packages. Redundancy analysis (RDA) was performed
in the R 'vegan' package to examine the relationship among frequencies of ASVs, samples and soil
variables, which were selected using 'stepAIC' in R. Variance partitioning analysis (VPA) was used to
determine the contributions of soil properties, sample location, and yield, as well as interactions among
the variation in a microbial community with hellinger-transformed data. The predictors of selected soil
properties for explaining the pear yield were identified by random forest regression analysis (Boulesteix,
et al., 2012). The significance of each predictor in the response variables was assessed with the
'rfPermute' package with 1000 permutations based on 1000 trees. Structural equation modelling was
applied to evaluate relative contributions of soil chemical properties and bacterial community to pear
yield (Schermelleh-Engel et al., 2003). The conceptual SEM fitness was examined on the basis of a non-
significant chi-square test ($P > 0.05$) and the goodness-of-fit index (GFI). Model was fitted using the
'lavaan' package in R (Rosseel, 2012).
**3    Results**





### 3.1 Overview of sequencing data

In total, 1,622,858 16S rRNA sequences were retained after quality control and a total of 9,394 ASVs were obtained for the 16S rRNA gene sequences based on 97% similarity. Among the total 16S rRNA gene sequences, 159 ASVs with 74,372 sequences were classified as Archaea while 9,235 ASVs with 1,548,486 sequences were identified as Bacteria. Among Bacteria, Acidobacteria, Proteobacteria, Chloroflexi, Planctomycetes and Actinobacteria were the most abundant phyla (Fig. S1).

### 3.2 Bacterial abundances and community compositions

Yield-invigorating orchards together displayed significantly higher abundances of total bacteria than that in co-located yield-debilitating orchards based on real time PCR result (Fig. 2B). Meanwhile, bacterial community compositions at the ASV level were significantly correlated to pear yield ($r = 0.460$, $p = 0.001$) (Fig. 1C).

**Fig. 1 here**

PCoA based on Bray-Curtis distance matrices clearly revealed treatment-based differences in bacterial community compositions (Fig. 2A). Six distinct groups representing samples from different locations (FX, GC, SN, TS, TX and ZJ) were obviously separated and confirmed by PERMANOVA test ($F = 14.9$, $P = 0.001$). At each location, soil bacterial community composition in yield-invigorating orchards was significantly separated from that in co-located yield-debilitating orchards, which was also confirmed by PERMANOVA test ($F = 3.6$, $P = 0.001$). Although only the Shannon diversity in yield-invigorating orchards from GC and ZJ was significantly higher than that in co-located yield-debilitating orchards (Fig. S2), the mean alpha diversity indices of Chao and Shannon in all yield-invigorating orchards were significantly higher than those in all yield-debilitating orchards based on the paired Wilcoxon test (Fig. 2B).

**Fig. 2 here**

The Venn diagram showed that 4540 ASVs occupying over 90% of total sequences were shared between yield-invigorating and -debilitating orchards (Fig. 2C). Among these shared ASVs, the fold changes larger than 2 of ASVs in yield-invigorating compared to yield-debilitating orchards were potentially linked to yield improvement. Surprisingly, none of these ASVs potentially linked to yield improvement were shared among six separated collocated orchards (Fig. S3). A total of 2546 unique ASVs with 53,222 sequences were found in all yield-invigorating orchards and 2308 unique ASVs with 44,389 sequences were observed in all yield-invigorating orchards, among which almost 70% of total



ASVs were shared among these unique ASVs between yield-invigorating orchards and -debilitating
orchards. However, no shared unique ASVs were found among six separately located orchards. The
affiliation of unique and shared ASVs at the phylum level exhibited that the Proteobacteria,
Planctomycetes, Chloroflexi, Acidobacteria and Actinobacteria were the top five phyla (Fig. 2D).
At the phylum level, the relative abundances of bacterial dominant phyla varied across the location
and orchard yield condition (Fig. 3A). Proteobacteria, Acidobacteria, Actinobacteria, Chloroflexi and
Planctomycetes were the top five abundant phyla. The mean abundance of Chloroflexi and
Planctomycetes was significantly higher while Firmicutes was significantly lower in yield-invigorating
orchards compared to yield-debilitating orchards based on Wilcoxon test (Fig. 3A, Fig. S4).
At a finer resolution, 967 genera were observed for all soil samples, among which 299 genera
appeared in more than half of soil samples in yield-invigorating or -debilitating orchards. However, only
34 genera displayed significant differences between yield-invigorating or -debilitating orchard soils
based on STAMP analysis (Fig. 3B). Interestingly, *Ornatilinea*, *Ktedonobacter*, *Longilinea*, belonging
to Chloroflexi, were significantly enriched in yield-invigorating orchard soils. *Gimesia* in
Planctomycetes and *Arenimonas* in Proteobacteria showed significantly higher relative abundances in
yield-invigorating orchard soils than in yield-debilitating orchard soils.

**Fig. 3 here**

3.3  Co-occurrence patterns of bacterial community
The phylogenetic molecular ecological networks were constructed using the random matrix theory-based
approach to explore the organization of bacterial communities in yield-invigorating (YI) or yield-
debilitating (YD) soil samples. After filtering ASVs that occurred in less than half of soil samples, 591
ASVs for yield-invigorating samples and 485 ASVs for yield-debilitating samples were used to construct
the networks. The YI network contained 302 nodes, 448 edges, and 11 larger modules (> 5 nodes), with
an average connectivity (avgK) of 2.967, average path distance of 5.494 and clustering coefficient
(avgCC) of 0.152, while the values in the YD network were 235, 334, 9, 2.843, 6.232 and 0.131,
respectively (Fig. 4A, Table S2). The module eigengene network analysis revealed a difference in the
higher-order organization between the two networks. Notably, the node composition was substantially
different between the two networks as the relative abundances of dominant phyla were obviously
different among different modules (Fig. 4B and C). A higher proportion of nodes in the module of YI
network was unique. ASVs affiliated to Acidobacteria, Chloroflexi, Proteobacteria, Actinobacteria, and



Planctomycetes within the unique modules (M9, M10 and M11) were observed in the YI versus YD
network.

Analysis using the threshold values of $Zi$ and $Pi$ showed that majority of nodes from both networks

were categorized as peripherals that had only a few links and almost always linked to the nodes within
their own modules (Fig. 4D). Although only three nodes affiliated to Acidobacteria were categorized as
module hubs in the YI network, seven nodes belonging to Acidobacteria, Actinobacteria and
Proteobacteria were categorized as module hubs in the YD network. Interestingly, four nodes including
*Longilinea* species from Chloroflexi in the YI network whereas only one node in the YD network was
categorized as module connectors (Table S3).

**Fig. 4 here**

3.4  Relationships between soil chemical properties and microbial community composition
Soil chemical properties differed significantly among the locations and orchard yield types (Table S4).
Together, yield-invigorating orchards exhibited a significantly higher content of soil organic matter (OM)
compared to yield-debilitating orchards based on the Wilcoxon test. Soil chemical properties were
significantly correlated to the bacterial community compositions (Mantel: r = 0.803, $p$ = 0.001). Soil
chemical properties, location, and orchard explained 44.9% of the observed variation, leaving 55.1% of
the variation unexplained for bacterial community composition based on VPA analysis (Fig. 5A).
Variation in the community composition was largely explained by soil properties (42.3%), and was also
influenced by locations and orchard yield types.

After forward stepwise selection, the module including soil OM, TN, alkaline N, AP and AK,

available calcium (Ca), copper (Cu) and manganese (Mn) explained the majority of the variation in
bacterial community composition (Fig. 5B). As evidenced by the RDA vectors, OM was among the most
important soil properties in shaping bacterial community composition. Random forest analysis showed
that contents of soil Mn, OM and Ca were the top parameters for predicting the orchard yield (Fig. 5C).
Furthermore, soil OM was also significantly correlated with bacterial communities as revealed by Mantel
test (Fig. 5D, Table S5).

**Fig. 5 here**

3.5  Relationships of soil chemical and microbial indicators with orchard yield
Soil OM as potentially key soil chemical properties and bacterial alpha diversity, beta diversity and
relative abundance of Chloroflexi and Planctomycetes as potentially key microbial indicators in



determining pear yield were used to construct a model to explain yield improvement. Final structural
equation modelling (path analysis) (Fig. 6) showed that the strongest driver explaining yield
improvement was beta diversity of bacterial community (PCoA) (r = 0.959, $P$ < 0.001), which was
positively affected by content of soil OM (r = 0.843, $P$< 0.001). Alpha diversity (Chao) of bacterial
community also determined yield improvement to a large extent (r = 0.542, $P$ = 0.009). However, alpha
diversity was not significantly correlated with content of soil OM.

**Fig. 6 here**

### 307    4    Discussion

Although pear is among the most important fruits worldwide, soil microbial communities in pear
orchards have been largely under-investigated (Huang et al., 2019). The present study attempts to
decipher the bacterial community linked to high-yield production of pear. Our results based on Mantel
analysis suggested a directly significant correlation between bacterial community and pear yield.
Microbial characteristics responding to yield promotion have repeatedly been observed on several crops
depending on single experimental site (Zhong et al., 2020; Qiao et al., 2019; Shen et al., 2013). It
remained unclear, however, whether these distinctions are ubiquitous at a large-scale. By comparing
multiple co-located yield-invigorating and -debilitating orchards, we demonstrate that high-yielding pear
production soils harbored shared bacterial communities with high diversity, significantly enriched
indigenous microbes and well-organized interaction network, which was triggered by soil organic matter.
Here we discussed these main results and potential mechanisms in detail.

Microbial diversity is critical to soil ecosystems in maintaining the integrity, function and long-term

sustainability (Kennedy and Smith, 1995). Higher soil biodiversity is considered to be linked to a more
stable system and enhance the combination of vital microbial functions and processes (Cardinale et al.,
2006; Bell et al., 2005). In line with a previous report that crop yield was correlated to the soil bacterial
diversity (Zhao et al., 2014), greater diversity of bacterial community in yield-invigorating soils was
observed in the present study. Our results indicate that higher microbial diversity may result in a more
stable agroecosystem, contributing to sustainable pear production.

In this study, we found that Proteobacteria, Acidobacteria, Actinobacteria, Planctomycetes and

Chloroflexi were the top abundant phyla. This result roughly agreed with previous studies showing that
Proteobacteria, Acidobacteria and Actinobacteria are usually dominant bacterial taxa in agricultural soils
(Xun et al., 2019; Dai et al., 2018), while Planctomycetes and Chloroflexi exhibit an unexpectedly high



relative abundance in rice cropped soil (Edwards et al., 2015) and sandy loam soil (Pathan et al., 2021).
The highest relative abundance of Proteobacteria was probably explained by the fact that Proteobacteria
are considered as copiotrophic bacteria and always flourish in soils with large amounts of available
nutrients (Fierer et al., 2007).

Moreover, a significantly higher abundance of Planctomycetes and Chloroflexi was observed in

yield-invigorating orchards, indicating that Planctomycetes and Chloroflexi may be responsible for pear
yield-improvement. There is no direct evidence showing that Planctomycetes could improve plant
growth, however, Planctomycetes has been reported to be involved in the soil biological processes such
as ammoxidation, carbohydrate and polysaccharide metabolic (Fuerst, 2017). This implies that
Planctomycetes may promote plant production through improving soil biological fertility. Chloroflexi is
a facultative anaerobic phylum including autotrophic, heterotrophic and mixotrophic taxa (Speirs et al.,
2019). Considering that soil amended with organic fertilizer could enhance the soil water holding
capacity, the yield-invigorating soils with more organic material input have a higher soil moisture content,
especially after irrigation, probably leading to the enrichment of Chloroflexi in soil. Furthermore, it has
been well documented that Chloroflexi could grow well in drought conditions (Ullah et al., 2019),
implying that yield-invigorating soils with a higher relative abundance of Chloroflexi may exhibit
excellent resistance to environmental stress to support sustainable crop production.

Network analysis is a systems-level method to explore interactions within an ecosystem that cannot

be directly observed through co-occurrence analysis (Fath et al., 2007). Similar to the food web network
analyses in macro ecosystems, microorganisms also form complex interactions with other species (Faust
and Raes, 2012) and have been widely investigated to explore the linkage of microbial network with soil
function, such as nutrient supply (Fan et al., 2021) and disease suppression (Lu et al., 2013). Overall, in
line with previous findings (Hu et al., 2020), the topological properties of the constructed networks,
including connectivity, average clustering coefficients, average degree distance, and modularity indicate
that these networks are scale-free, modular and "small world". Our comparative network analysis
indicated that microbial co-occurrence patterns in soils were correlated to pear production. As a meta-
module is usually considered as a group of modules functionally interrelated (Langfelder and Horvath,
2007), a greater number of meta-modules were identified in the network constructed from yield-
invigorating soils, suggesting that a greater number of network nodes in the yield-invigorating soils were
functionally interrelated than those in the yield-debilitating soils. A majority of nodes in the meta-





modules were not shared between yield-invigorating and -debilitating networks, indicating basal shifts
in network architecture during pear production with contrasting yield performance.

Furthermore, a higher proportion of negative interactions to positive interactions were identified in

the network constructed from yield-invigorating network than the yield-debilitating network. Our results
indicated that stronger resource competitions in yield-invigorating soils, which means that the soil co-
occurrence network was more stable to maintain soil ecosystem function (Coyte et al., 2015). In this
study, three module connectors and three module hubs were identified as potentially key taxa in the yield-
invigorating network. Interestingly, among those key species, ASV357 affiliated to *Longilinea*,
belonging to the Chloroflexi, was recognized as a key phylum in improving pear yield. Similarly,
Chloroflexi was reported to be key-stone taxa in the constructed network from agricultural soils with 40-
years fertilization (Fan et al., 2021). Chloroflexi play key roles in manipulating soil microbiome probably
due to that Chloroflexi could participate in degrading plant compounds to create more niches via
pathways for the degradation of starch, pyrogallol, cellulose, and longchain sugars, as it is positively
correlated with genes for amino sugars, sugar alcohols and simple carbohydrate metabolic pathways
(Hug et al., 2013).

Soil pH is generally recognized as the main driver in the assembly of bacterial community,

especially in the studies related to geographic distribution of microorganisms (Fierer and Jackson, 2006).
Soil pH varying across a wide range allows insights into the relationships between pH and soil bacterial
communities in those researches. Therefore, we speculated that there are important factors other than pH
in shaping soil bacterial communities in our study, given that soil pH only ranged within two units. In
this study, a significantly higher content of soil organic matter was observed in yield-invigorating
orchards, demonstrating that soil organic matter can also drive the assembly of bacterial community.
Consensus is emerging that microbial materials are an important constituent of soil organic matter
(Kallenbach et al., 2016), a higher content of soil organic matter usually supports a more diverse
microbial community, which participate in almost all soil biological processes (Fierer, 2017).

Structural equation modelling approach has been widely used to decipher keystone indicators

associated with soil function and crop production in agroecosystems (Jiang et al., 2020; Chen et al., 2019).
In the present study, we observed that soil organic matter, beta diversity of bacterial community, and
network connector were key indicators in supporting high-yield pear production based on the structural
equation modelling results. Therefore, we proposed that yield-invigorating soils harbour unique bacterial



communities that could improve soil biological fertility, which could be driven by soil organic matter and manipulated by keystone species (Chloroflexi) through altering the bacterial interactions.

**5    Conclusions**

In conclusion, by comparing six paired-located orchards, our results demonstrated that yield-invigorating soils showed a higher content of organic matter and harboured unique bacterial community with greater diversity than yield-debilitating soils. We further highlight that Chloroflexi was significantly enriched and served as a keystone taxon in manipulating the interaction of bacterial community in yield-invigorating soils. These findings help elucidate the role of soil microbiome in maintaining crop production and factors controlling the assembly of soil microbiome. Such knowledge is a first step toward harnessing soil microbiome in support of sustainable agroecosystems.

**Data availability.** Raw amplicon sequencing data for each sample used in this study was deposited at the National Center for Biotechnology Information (NCBI) in the FASTQ format and is available under the accession number PRJNA749397. Other data that support the findings of this study are available on request from the corresponding author (Xiaomei Ye).

**Authors' contributions**

L. Wang: performed all experiments; L. Wang, X. Ye, and Z. Shen: designed the study, and wrote the majority of the manuscript; L. Wang and Z. Shen and C. Tao: analyzed the data; H. Hu, J. Du, Y. Xi, J. Lin, and D. Chen: participated in the design of the study, provided comments and edited the manuscript. The authors read and approved the final manuscript.

**Competing interests.** The authors declare that they have no conflict of interest.

**Disclaimer.** Publisher's note: Copernicus Publications remains neutral with regard to jurisdictional claims in published maps and institutional affiliations.

**Acknowledgements.** We sincerely thank all those who have assisted me with any part of this paper and all pear orchards owners for providing access to the soil sampling.



**Financial support.** This research was supported by Jiangsu Agricultural Science and Technology
Innovation Fund (CX(19)3094) and National Natural Science Foundation of China (31801842 and

42090065).


**Supplementary data**
Supplementary figures and tables to this article can be found in the supplemental material.

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



**Figure legends**
**Fig. 1 Distribution of field sites, quantitation of the abundance of bacteria population, and linkage**
**of microbial composition to pear yield.** (A) Map showing the sites of six pair-located orchards sampled
in this study. (B) Violin plot showing the abundance of total bacteria for all selected orchards. * indicates
a significant difference between yield-invigorating (YI) and yield-debilitating (YD) orchards based on
Wilcoxon tests ($p < 0.05$). (C) Correlation plot showing the relationship of microbial composition and
yield based on braycurtis distances calculated by Mantel test.

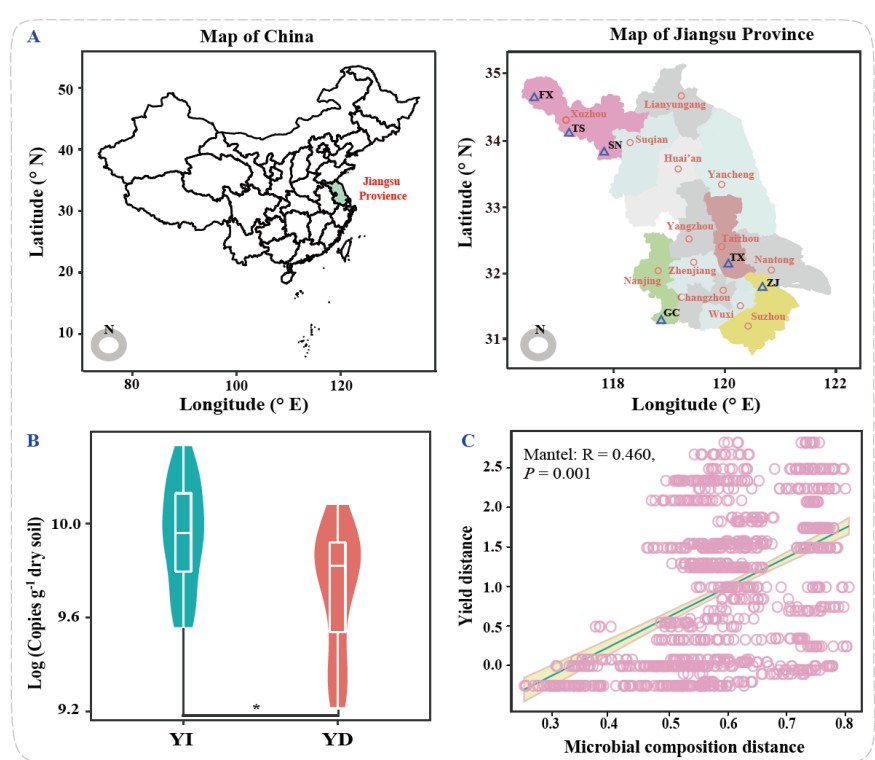





**Fig. 2 Overview of bacterial composition and alpha diversity.** (A) Principal Coordinates Analysis
(PCoA) plot displaying the bacterial community composition calculated based on braycurtis distances.
(B) Violin plot showing the alpha diversity indices (Chao and Shannon) for all selected orchards. *
indicates a significant difference between yield-invigorating (YI) and yield-debilitating (YD) orchards
based on Wilcoxon tests ($p < 0.05$). (C) Venn plot depicting the unique and shared bacterial ASVs
between yield-invigorating (YI) and yield-debilitating (YD) orchards at ASV and sequence insights.
Uni. YI and Uni. YD represent unique ASVs or sequences in the YI or YD soils while Shared represent
shared ASVs or sequences between the YI and YD soils. (D) Heatmap displaying the composition of
unique and shared ASVs at phylum level in YI and YD soils. Numbers in the cell represent the number
of ASVs affiliated to that phylum.

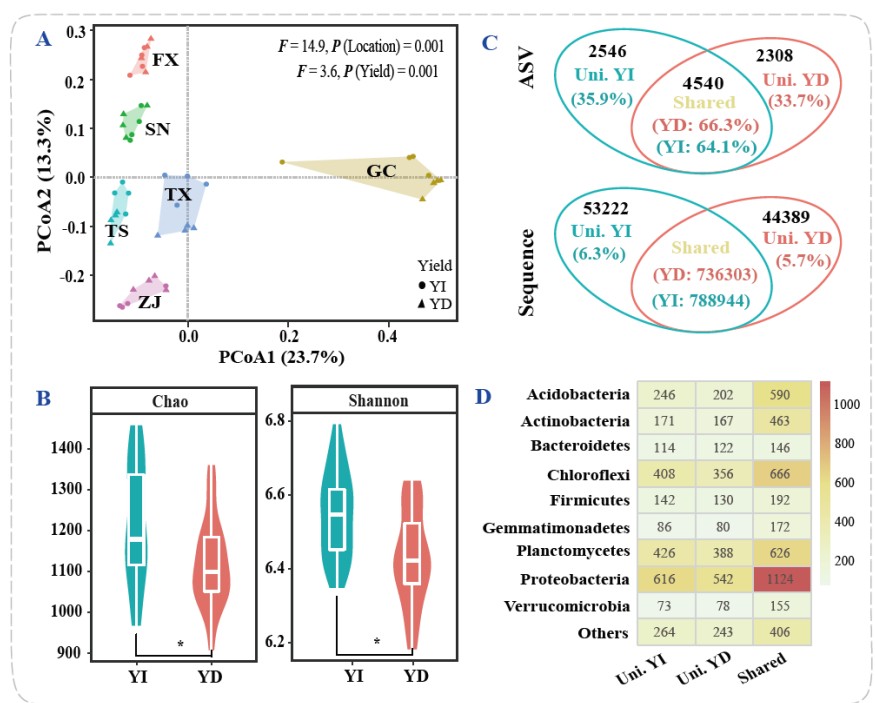







**Fig. 3 Key taxonomic groups in distinguishing yield-invigorating (YI) and yield-debilitating (YD) orchards.** (A) Stacked bar chart (left panel) showing dominant phyla affiliation in YI and YD soils for six pair-located sites while horizontal histogram (right panel) depicting relative changes of dominant phyla in YI soils compared to those in YD soils. (B) Genus-level taxonomic analysis of bacterial sequences obtained from yield-invigorating (YI) and yield-debilitating (YD) orchards using the STAMP software. Cyan bars represent the YI soils and pink bars represent the YD soils.

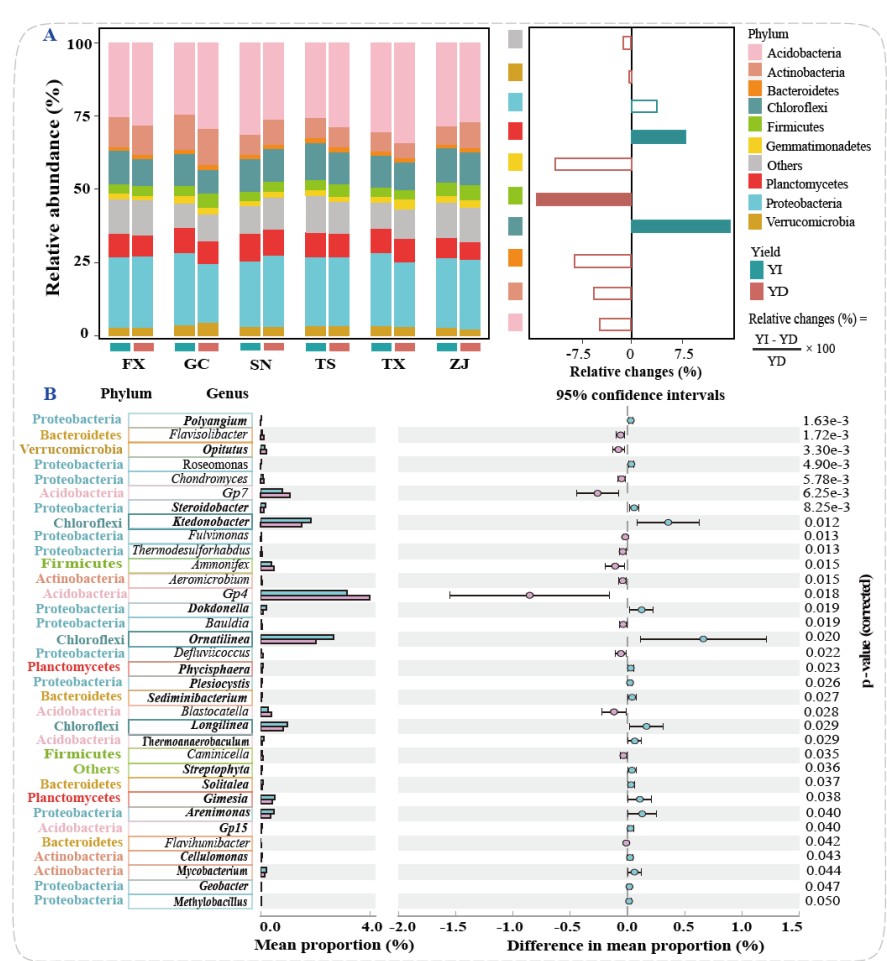

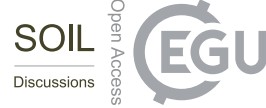

**Fig. 4 Co-occurrence networks of bacterial community and identified keystone taxa in**
**distinguishing yield-invigorating (YI) and yield-debilitating (YD) orchards.** (A) An overview of
microbial phylogenetic molecular ecological networks constructed from YI and YD soils. Line with blue
color indicates positive correlations whereas lines with red color signifies negative correlations in each
network. Modules containing larger than five nodes in the networks are labeled with corresponding letter
followed by a number. Circular node colors indicate different bacterial phyla. (B) Bubble graph showing
the relative abundance of nodes in each module within each network at the phylum level. (C) Venn plot
depicting the unique and shared bacterial ASVs between two networks construed from YI and YD soils.
Left panel is plotted based on the original nodes used in building network while right panel is plotted
based on the nodes from modules. Uni. YI and Uni. YD represent unique ASVs in the YI or YD networks
while Shared represent shared ASVs between the YI and YD networks. (D) Zi-Pi plot showing the
distribution of nodes based on their topological roles. The threshold values of Zi and Pi for categorizing
OTUs were 2.5 and 0.62 respectively. Node colors indicate different bacterial phyla and node size
represent the relative abundance in each network.

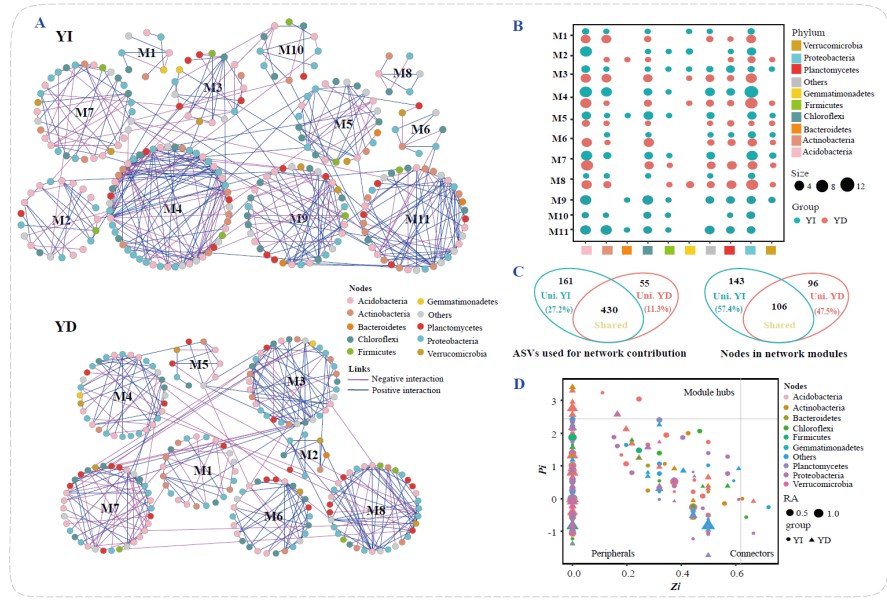





**Fig. 5 Relationships among bacterial community, soil edaphic factors and pear yield.** (A) Variance
partitioning analysis (VPA) map of the effects of soil edaphic properties, sample locations, pear yield
and interactions of these factors on the bacterial community. (B) Redundancy analysis (RDA) plot
showing the relationships among all assigned bacterial ASVs and measured soil edaphic properties for
all soils after stepwise selection. (C) Random forest mean predictor importance of selected soil edaphic
properties used in the as drivers in predicting the pear yield. Red bar indicates that the given predictor
is significant while black bar indicates that the given predictor is non-significant. (D) Correlation plot
showing the relationship of microbial composition and soil organic matter based on braycurtis
distances calculated by Mantel test.

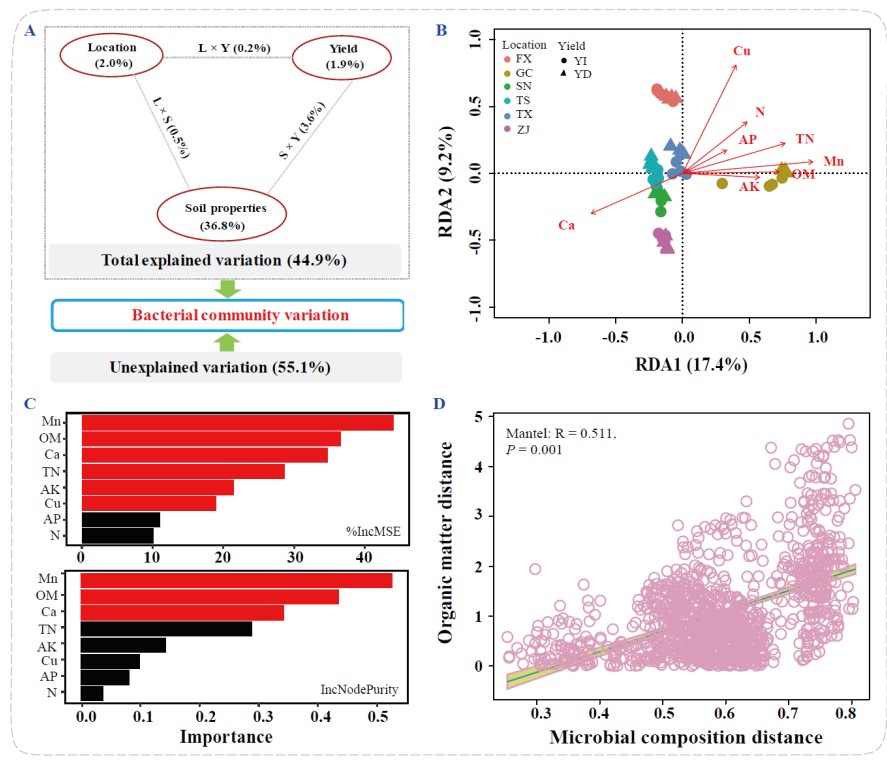





**Fig. 6 Structural equation modeling (SEM) describing the biotic and abiotic factors in affecting**
**the crop production.** Structural equation model was built incorporating soil organic matter, microbial
biomass, beta diversity (PCoA), key taxa, network hubs including module hubs and network
connectors, and yield. The path analysis numbers adjacent to arrows indicate the relationship's effect
size and the associated bootstrap *P*-value. Cyan and red arrows indicate positive and negative
relationships, respectively. Paths with non-significant coefficients are presented as gray lines.

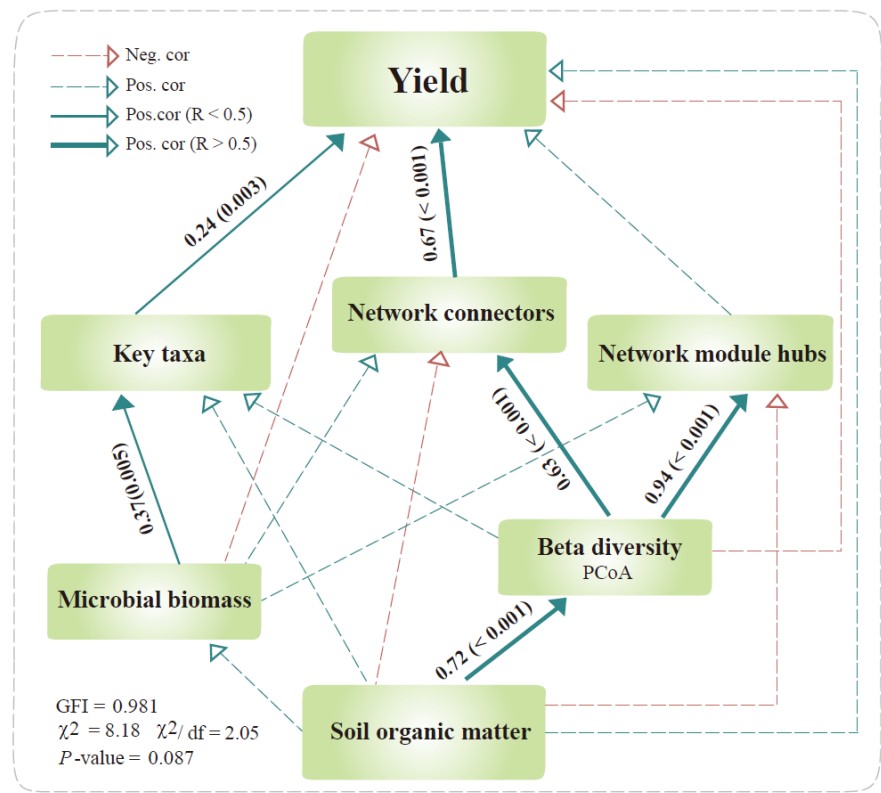







