# Peer review of "Soil bacterial community triggered by organic matter inputs"

_SOIL, 2021_

## Author Comment (AC1)

**Comments**

**Review1:**

The authors of the study "Soil bacterial community triggered by organic matter inputs supports a highyielding pear production" addressed an important and interesting topic by searching for specific characteristics and members of the soil microbiome that can be linked to soil fertility, in this case expressed as the yield of pears. The authors present a well-designed study with appropriate and sophisticated methods. The overall presentation of the study is well-done. I have some minor issues and one major issue, which needs consideration before publication.

[1] My major issue is related to the causality of the relation between the microbiome and soil fertility/pear yield. This becomes mostly obvious in the discussion, but also elsewhere in the manuscript. In this context, it is important to distinguish between direct conclusions that can be drawn from your results and speculations. It seems to me that SOM is the main driver as mentioned in L317. However, in the following discussion, the picture arises, that the microbiome is the main driver without showing clear mechanistic evidence from the data. It is therefore important to keep the right order of effects in mind, which would be, from my point of view, SOM change ==> microbiome change ==> fertility. An additional benefit of the more complex networks for soil fertility could be the resistance against stress. However, this was not tested in this study and could only be hypothesized. In such a case, there would be a direct mechanistic link between fertility and the microbiome. Overall, I suggest that the authors are more careful with what they conclude from their data. The approach to look for some general characteristics of the microbiome and its correlation with yield across different sites is interesting enough in itself.

**Response:** Thanks for your comment. First, combined the comments from another reviewer, we revised our result section. We introduced our result as your suggestion in the order of soil properties, microbiome, and their relationships. Second, we total agreed with your opinion that we lack of experimental evidence that the agroecosystem is more stable. Therefore, we revised it as "Hence we infer that higher microbial diversity may result in a more productive agroecosystem, contributing to sustainable pear production.". We also checked other parts of discussion to improve the Discussion section.

**Specific comments**

[2] L36 I suggest to write this more carefully. Factor suggests that there is a mechanistic link between e.g. beta diversity and productivity of the orchard. However, as long as there are no ideas about specific mechanisms, it might be that this is simply a non-causal correlation. For example, both might be primarily driven by SOC without direct mechanistic link. However, this is of course a valuable result and I would write it in a way that points towards the need to underpin this relation with mechanistic studies.

**Response:** Thanks for your warm suggestion. Indeed, this result was a statistic correlation and a further experiment needed to validate this correlation. Therefore, we revised this sentence as following: "Structural equation modelling showed that soil organic matter, beta diversity of bacterial community, and network connector (Chloroflexota) were identified as potential key factors in explaining the high-yield pear production.".

[3] L80-87 I suggest to strongly reduce this paragraph. The details about the importance of pears in China are not so important for this study. It is enough to know that pears are an important fruit and that improving the productivity is desirable.

**Response:** Thanks for your warm suggestion. We have reduced these sentences as following: "Sucui No. 1' pear, is an early-maturing variety bred by the Jiangsu Academy of Agricultural Sciences, China, and has been popularly cultivated in Eastern and Central China, due to the advantages including easy to produce, adaptable to the environment, and has good quality and high economic benefits (Lin et al., 2013).".

[4] L95 "...properties of the study sites were compared..."

Response: Thanks for your warm suggestion. We have revised it.

[5] L155 What does "roughly according" mean? I suggest to shortly present the methodological deviations from the reference.

**Response:** Thanks for your warm suggestion. We sequencing method was similar according to the cited method. Therefore, we delete the word of "roughly".

[6] Fig 2B I miss a label on the y axis, something like "bacterial abundance".

Response: Thanks for your warm suggestion. We have added the missing labels.

[7] L234-236 I don't understand the message of this sentence. Please revise this sentence.

**Response:** Thanks for your warm suggestion. We have revised this sentence as following in the revised manuscript: "Among these shared ASVs, the fold changes larger than 2 of ASVs in yield-invigorating compared to yield-debilitating orchards were recognized as potential responders linking to yield improvement."

[8] L260 Please provide some reasons for filtering the ASVs. With this analysis you want to characterize the different communities, but you exclude part of the communities.

**Response:** Thanks for your warm suggestion. Here we filtered the ASVs appeared in less than half of soil samples. These ASVs appeared less than half of sampled soils may largely depended on specific location due to soil variational physic-chemical properties. Here we revised the sentence as following "Given the large number of rare taxa that are specific to certain locations, ASVs that occurred in less than half of soil samples were filtered, which resulting in 591 and 485 ASVs for YI and YD samples respectively, before networks constructed."

[9] L 272 What are the Zi and Pi values and what do they mean. This needs to be explained for readers that are not familiar with the applied statistics.

**Response:** Thanks for your warm suggestion. Here Zi means the within-module connectivity and Pi refers to among-module connectivity. We have added this introduction in the manuscript.

**[10] L286 VPA analysis?**

Response: Thanks for your warm suggestion. We have revised it as "VPA result".

[11] Fig 5C Please explain the meaning of the two graphs in 5C.

**Response:** Thanks for your warm suggestion. We have added these explanations in the figure legend of Fig. 5.

[12] L324-325 You cannot say that these agroecosystems are more stable, because you did not investigate the stability, i.e. the response to stress. Please revise this sentence accordingly.

**Response:** Thanks for your warm suggestion. We have revised it as "Hence we infer that higher microbial diversity may result in a more productive agroecosystem, contributing to sustainable pear production." in the revised manuscript.

[13] L335 I wouldn't say "responsible". In the following sentences, you do not provide evidence from the literature, that there might be a direct mechanistic link between the abundance of these phyla and yield productivity. "To be involved in biological processes..." is by far not specific enough to provide such a mechanistic link. You also write that Chloroflexi have taxa with different ecological traits. However, since you did not evaluate, whether specific taxa with specific traits are dominating the abundance of Chloroflexi in your soils, your conclusion are based speculation and not on sound literature knowledge.

Response: Thanks for your warm suggestion. We have revised "responsible for" as "associated with".

**[14] L338 Which processes?**

**Response:** Thanks for your warm suggestion. We mentioned it in the revised manuscript as following: "Planctomycetes has been reported to be involved in the soil biological processes such as ammoxidation, carbohydrate and polysaccharide metabolic (Fuerst, 2017)."

[15] L354 What does "scale-free, modular and "small world" exactly mean? Please provide some more background on this.

**Response:** Thanks for your warm suggestion. We have provided the background on this in the revised manuscript as following: "Overall, in line with previous findings (Hu et al., 2020), the topological properties of the constructed networks, including connectivity, average clustering coefficients, average degree distance, and modularity indicate that these networks are scale-free, modular and small world. In short, a scale-free network represents that a network whose connectivity follows a power law, and most of nodes have only a few connections with other nodes. Meanwhile, a small-world network is the network in which most nodes are not neighbors of one another, but most nodes can be reached by a few paths. Modularity is a fundamental characteristic of biological network as a module in the network is a group of nodes that are highly connected within the group, but very few connections outside the group."

[16] L364 "...indicated stronger..."

Response: Thanks for your warm suggestion. We have revised it as your suggestion.

[17] L365 Which study do you address with "this study"? Yours or Coyte et al. 2015?

**Response:** Thanks for your warm suggestion. Here is the result of our study. Therefore, we have revised it as "In our study".

[18] L369 Usually, most of the agricultural ecosystems are fertilized. It would be, therefore, interesting to know whether this was an organically fertilized soil.

**Response:** Thanks for your warm suggestion. The soils in the cited study (Fan et al., 2021) was fertilized with different regimes including mineral fertilizer or chemical fertilizer plus organic fertilizer. The network was constructed by all treatments. It is hard to say the soil was organically fertilized soil. Therefore, we only mention agricultural soil here.

[19] L370 Do the Chloroflexi really "manipulate" the microbiome, i.e. intending to actively change the microbiome? Or is it that they merely affect the microbiome by their degradation of polymers? Maybe, they are "only" representatives for the change in the microbiome, which was induced by organic fertilization, i.e. they are not active players but passive responders? Please elaborate more on such questions, because it seems to me important, that your nice results are properly discussed based on direct conclusions from your results and not on speculation.

**Response:** Thanks for your nice suggestion. We agreed all the ways you suggested here. And a lot of following experiments need to be conducted to answer these questions. Chloroflexota could degrade soil organic matter that may produce more nutrients to soil microbes, which may stimulate or manipulate soil microbiome. However, we lack of direct evidence that this is the Based on the network result, we can infer that Chloroflexota (a new name of the Chloroflexi) may paly key roles in manipulating soil microbiome since the does belonging to the phylum was identified as module hub in YI network. Therefore, we revised this sentence as following: "Chloroflexota play key roles in connecting network nodes of soil microbiome probably due to that Chloroflexota could participate in degrading plant compounds to create more nutrients via pathways for the degradation of starch, cellulose, and longchain

sugars, as it is positively correlated with genes for amino sugars, sugar alcohols and simple carbohydrate metabolic pathways (Hug et al., 2013)."

[20] L372 What is the ecological relevance of pyrogallol?

**Response:** Thanks for your warm suggestion. Here we tend to show that plant compounds including pyrogallol can be biodegraded by the Chloroflexi members. However, to eliminate misunderstanding here, we have deleted this work in the revised manuscript.

[21] L375-391 These two paragraphs are disconnected to the line of argumentation, which was presented before. Please integrate these two paragraphs better into your discussion.

**Response:** Thanks for your warm suggestion. Combined the comment from another review, we have revised this two paragraphs as following:

"In this study, a significantly higher content of soil organic matter was observed in yieldinvigorating orchards, demonstrating that soil organic matter could drive the assembly of bacterial community. Consensus is emerging that microbial residues are an important constituent of soil organic matter (Kallenbach et al., 2016), , which participate in almost all soil biological processes (Fierer, 2017). the Despite the quality of soil organic matter was not evaluated in this study, the quality of soil organic matter was associated with the diversity of microbial community (Ding et al., 2015), which implies more attentions should be paid to illustrate the relationship between the quality of soil organic matter and microbial community in our future work.

Structural equation modelling approach has been widely used to decipher keystone indicators associated with soil function and crop production in agroecosystems (Jiang et al., 2020; Chen et al., 2019). In the present study, we observed that soil organic matter, beta diversity of bacterial community, and network connector were key indicators in supporting high-yield pear production based on the structural equation modelling results. Worth to mention, soil organic matter was not directly linked to the yield in the constructed model, indicating that soil organic matter maintain the high-yielding pear production probably via the indirect ways. Therefore, we proposed that yield-invigorating soils harbour unique bacterial communities that may improve soil biological fertility, which could be driven by soil organic matter and manipulated by keystone species (Chloroflexota) through altering the bacterial interactions.".

[22] L382 better "microbial residues"

Response: Thanks for your warm suggestion. We have revised it as your suggestion.

[23] Conclusion The conclusion is more a repetition of the main results than a presentation of conclusion related to the larger challenge.

**Response:** Thanks for your warm suggestion. We have revised it as following: "In conclusion, yieldinvigorating soils displayed a higher content of organic matter and harboured unique bacterial community with greater diversity than yield-debilitating soils. Further Chloroflexota was significantly enriched and identified as a potential keystone taxon in manipulating the interaction of bacterial community in yieldinvigorating soils. These findings indicated that soil organic matter triggered the assembly of soil microbiome, which both participated in maintaining crop production. Such knowledge is a first step toward harnessing soil microbiome in support of sustainable agroecosystems.".

**Review2:**

The authors of the manuscript soil-2021-95 "Soil microbial community triggered by organic matter inputs supports a high-yielding pear production" present a dataset based on 6 sites, each with a highyield and low-yield treatment, named "yield-invigorating" and "yield-debilitating". Four composite samples (composite of nine individual samples) were taken per treatment. According to the introduction, the authors assume the higher yield at "yield-invigorating" pear orchards to be associated to a unique microbial community, which in turn is affected by abiotic factors. The objectives of this study were to i) identify differences in taxonomic diversity and composition of the bacterial community between the treatments, and ii) determine abiotic factors shaping the bacterial community composition. The broader implication stated would be the targeted manipulation of soil bacterial composition in order to support higher pear production. I want to highlight the impressive variety of statistical methods and results. I also appreciate the data provided in the supplemental material, which provides additional transparency. However, this variety of methods causes a lengthy results part, which sometimes loses the focus on the research question. While at the same time, important information is missing in the M&M section (i.e. a literature based metamodel explaining the rationales behind each pathway, how the model was fitted). I have three major concerns:

[1] One major concern is the lack of a clear definition of the treatments "yield-invigorating" and "yield-debilitating", thus both terms remain vague from abstract to conclusions. In M&M it is stated, that yield-debilitating orchards received more chemical fertilizer under intensive management (unclear what intensive management refers to), while yield-invigorating orchards were "usually" amended with more organic fertilizer under integrated nutrient management. How did the NPK and OM inputs differ between the treatments. And how consistent were the treatments across the six sites? Is the fertilisation regime the only difference between the two treatments? The terms "yield-invigorating" and "yield-debilitating" indicate a trend in increased or reduced yield over time, however, Table S1 does only provide yield data for 2019, and yield is only given as average yield per plot and not given for each individual replicate, which results in a limited observation of 12 yields.

**Response:** Thanks for your nice comment. In the present study, yield-invigorating (YI) orchard displays a higher pear fruit yield whereas yield-debilitating (YD) orchard shows a lower pear fruit yield compared that in orchard under local common managements according to the farm record. These terms indicate a trend of increase or decrease yield over time. In the present study, the orchard was classified according to the field survey and farm record especially annual production. In 2019, when soils were sampled, the yield of each orchard was calculated based on the farm record. Unfortunately, the farmers did not record the yield of each replicate, they only record the yield for the whole orchard. Therefore, we used the average yield of the whole orchard to represent the yield of each replicate. That is the limitation of our study. Anyway, combing the comment from another review, we introduced what the YI and YD means in the Introduction section in the revised manuscript.

We firstly performed a field survey to choose the pair-located orchards for further analysis. To minimize the effects of microclimate at each site, only pair-located pear orchards with invigorating and debilitating yield and at similar growth stage were selected for this research. In total, six pair-located yield-invigorating and -debilitating pear orchards distributed in four cities of Jiangsu province, China, were selected in the main pear production areas. Although these paired orchards differ in site, plant density and cropping years, however, the YI and YD orchards were the same except the soils sampled from Zhangjiagang. According to the farm records and our survey, for these paired orchards within each site, fertilization regime probably was the most different part during the agricultural management. The YI orchards usually received more organic fertilizer whereas YD orchards received more chemical

fertilizer. We have added this information in the supplemental Table S2 and revised the concerned sentences in the revised manuscript.

[2] Furthermore, the conclusions drawn are very speculative, and correlation between distinct bacterial communities between treatments with yield (the main factor defining treatment) are interpreted as causal for higher yields. However, yield varies largely between the different sites, as do the bacterial communities. An increase in key taxa might be linked to treatment, thus fertilisation (which was indicated to vary between treatments), but does not necessarily cause higher yields. There is a lack on discussion on the direct effect of higher SOM on yield, plant growth and root exudation (which might also affect microbial community composition).

**Response:** Thanks for your warm suggestion. In this study, we found that YI and YD orchards differed in their soil chemical properties, especially the contents of soil organic matter, which leading to the variation of soil microbiome. And the manipulation of soil microbiome correlated to the improvement of pear yield. Based on the result of structural equation modelling, we observed that soil organic matter, beta diversity of bacterial community, and network connector were key indicators in supporting highyield pear production based on the structural equation modelling results. Worth to mention, soil organic matter was not directly linked to the yield in the constructed model, indicating that soil organic matter maintain the high-yielding pear production probably via the indirect ways. Therefore, we proposed that yield-invigorating soils harbour unique bacterial communities that may improve soil biological fertility, which could be driven by soil organic matter and manipulated by keystone species (Chloroflexota) through altering the bacterial interactions. In this study, we hypothesized that high input of organic fertilizer could improve soil structure and modify chemical properties, which leading to YI soils harbour unique bacterial communities that maintains the high-yielding pear production. We have re-organized our Result and Discussion sections to improve the writing of our manuscript.

[3] Third, statistical analyses miss to investigate important effects of SOM on yield, and of pH on microbial community composition. The structural equation model approach could promote valuable insights and help to disentangle direct and indirect effects of SOM on yield. However, this would need an inclusion of a direct path from SOM to yield. Furthermore, the authors assume that soil pH (major

factor influencing bacterial community) is not of importance in the study cited. However, including pH into the model would allow to prove this assumption. In general, the SEM is lacking the information how much of variation in yield is explained. From Fig. 5A it looks like yield explained ~8% of variation in bacterial community. Vice versa, this questions the conclusion that the bacterial community structure affects growth.

**Response:** Thanks for your comment. The  $R^2$  is 0.739 for this SEM and we have added it in the revised manuscript. The raw data used for SEM construction was selected because these index may play key roles in supporting a high-yielding pear production based on the previous result in the present study. In this study, we firstly used the VPA analysis to show soil chemical properties play key roles in determining soil microbial communities. Next we used RDA analysis and stepwise to select key soil chemical properties driving soil microbial communities. Meanwhile, we used random forest analysis to explore the key soil chemical properties in predicting pear yield. Combining all these results together, soil organic matter was recognized as most important factor in driving soil microbial community and determining pear yield. For soil chemical properties, soil organic matter was only the factor selected to build the SEM mainly due to the following reasons: 1) only OM was significant differed between YI and YD soils, 2) OM was top parameters for predicting the orchard yield, 3) OM within the module was identified as the top important soil property that determines the composition of bacterial community as evidenced by the RDA vectors. Therefore, soil pH or another factors were not shown in our manuscript when building the SEM. And the direct correlation between soil OM and pear yield was added in the revised manuscript. the detailed information about the parameters used for SEM construction was added in the revised manuscript. Following your suggestion, we also added a conceptual model in the supplemental material.

Worth to mention, the built model including soil chemical and properties may explain 73.9% variation of yield based on the SEM result. As for microbial community, the result of VPA showed soil chemical properties together may explain 36.8% variations of microbial community while location and yield only explain a small proportion. In our opinion, this probably location and yield were not direct factor in shaping the changes of soil microbial communities. They are mainly affected the composition of soil microbial communities first through changing soil properties. And our results are in line with to many previous VPA results that soil chemical properties explaining the most of the variations of

microbial communities (Wu et al., 2021, 161: 108374; Yang et al., 2017, 215: 756-765; Zhao et al., 2013, 67: 443-453). Anyway, we have added this point in our Discussion section.

Unfortunately, I cannot recommend publication in its current form and therefore propose a rejection. I hope the authors still take the time to consider my recommendations regarding the revision of this manuscript.

[4] Title The shown data does not allow to assume causality between distinct bacterial community upon organic fertiliser inputs and high yield. Therefore, the title should be revised.

**Response:** Thanks for your warm suggestion. We have revised the title as "Soil bacterial community triggered by organic matter inputs was associated with a high-yielding pear production".

**Abstract**

[5] e.g. L33 Try to write the abstract in a way that it is understandable to a broader audience, without using to many very terms specific to a certain method (meta-modules).

**Response:** Thanks for your nice reminder. Meta-module usually refers to a group of modules functionally interrelated as the original references (Langfelder and Horvath, 2007, Oldham et al., 2008). We have revised this sentence in the abstract as your suggestion.

[6] L37 I don't agree on conclusion that the factors presented are causal for higher yields.

**Response:** Thanks for your suggestion. Agreed with your opinion, there are lots of biotic and abiotic factors determines the pear yield. However, in our present study, we analyzed the soil physiochemical properties and microbial composition, and used different statistical methods to discover the potential key factors responding for yield improvement. Here is the result from structural equation modelling. We lack of a further-step experiment to explore if these factors were really the causal for higher yields. Anyway, our result could give a basic information about which index probably links to yield improvement. Therefore, we revised our manuscript as following: "Structural equation modelling showed that soil organic matter, beta diversity of bacterial community, and network connector (Chloroflexi) were key factors in explaining the high-yield pear production."

introduction

[7] L52 Consider adding a sentence on the role of fungi in perennial agroecosystems (e.g. orchards). **Response:** Thanks for your suggestion. We have added the information as following: "Fungi participate in decomposition of organic matter and deliver nutrients for plant growth (Frac et al., 2018), however, considering that bacteria are the most diverse and abundant group of microorganisms in soil, bacterial communities and their functions can be pivotal indicators for crop production in agroecosystems (van der Heijden et al., 2008)."

[8] L59 Which monoculture system does Lu et al. 2013 refer to? Be more precise.Response: Thanks for your suggestion. We have revised it as your suggestion.

[9] L64 I miss the logical link to LLs 62-63. Why is it desirable to identify indicators of bacterial community composition in response to high-yielding crop production?

**Response:** Thanks for your suggestion. There are many microbial indices that probably respond to yield improvement such as a higher microbial diversity, interaction patterns, and enrichment of beneficial microbes as we introduced in the L55-L62. However, the relative contributions of microbial diversity, interactions among community members, or enrichment of key taxa to crop production remain largely unknown. Therefore, we mentioned here that it is highly desirable to identify pivotal indicators of bacterial community composition in response to high-yielding crop production.

[10] L70 What does "forms" mean?

**Response:** Thanks for your suggestion. Here means the form of soil nutrient. We have revised the sentence as "forms and contents of soil nutrient".

[11] L75 Delete therefore. Is not logically linked to the sentence in LLs 73-75.**Response:** Thanks for your suggestion. We have deleted it.

[12] L75-L77 Be more precise. Do you mean, that it depends on the scale which chemical properties are related to bacterial community composition? Does this refer to your assumed absence of pH as relevant factor in your soils? (L378-L379)

**Response:** Thanks for your suggestion. Here we mean the driver of soil microbial community was depending on the scales of soil samples investigated. At a large distance even at continental level, the assemblage of soil microbiome was determined by soil pH. At small scale, especially in agricultural soils, the assemblage of soil microbiome was determined by many soil factors. We revised the sentence to be more precise as following: "Key soil chemical properties identified in controlling the distribution and abundance of bacterial community is largely depending on the geographical distributions of soils".

[13] L80-L87 Reduce paragraph on pear to one sentence.

**Response:** Thanks for your suggestion. We have revised it as following: "Sucui No. 1' pear, is an earlymaturing variety bred by the Jiangsu Academy of Agricultural Sciences, China, and has been popularly cultivated in Eastern and Central China, due to the advantages including easy to produce, adaptable to the environment, and has good quality and high economic benefits (Lin et al., 2013).".

[14] L88-L89 What do you mean by that? It is unclear. Revise. Plus Zorz et al. 2019 refer to lake microbiomes. Consider choosing a reference related to agroecosystems.

Response: Thanks for your suggestion. We have revised it as your suggestion.

[15] L93-L94 It is unclear from introduction, why yield-invigorating bacterial communities should differ or be unique. Revise introduction so that your hypothesis has a better fundament.

Response: Thanks for your suggestion. We have revised it as following:

"In this study, compared to local average yield, orchard showing higher pear yield production was recognized as yield-invigorating (YI) orchard while orchard displaying lower pear yield production was regard as yield-debilitating orchard (YD). After field surveys accomplished in 2019, six yield-invigorating and adjacent yield-debilitating pear orchards in total were selected for further analysis of soil chemical properties and microbiome. We hypothesized that high input of organic fertilizer could improve soil structure and modify chemical properties, which leading to YI soils harbor unique bacterial communities that maintains the high-yielding pear production. To address this, soil bacterial communities

and edaphic properties of the study sites were compared were compared to (1) decipher the differences of taxonomic diversity, and composition of the bacterial community, and (2) determine the contributions of environmental variables to the changes in the structure of bacterial communities."

**methods**

[16] L116-L121 I miss a clear definition of the treatments. What does it mean, that yield-invigorating orchards were "usually" amended with more organic fertiliser. Did yield-debilitating orchards also receive organic fertiliser sometimes? Was the fertilisation regime/ the treatments consistent over all six sites? Table S1 does not provide "detailed information". It just provides information stated in L103. Could you provide detailed information on the amount of NPK and organic fertiliser inputs for each site? **Response:** Thanks for your suggestion. For both yield-invigorating and yield-debilitating orchards, chemical fertilizer and organic fertilizer were applied. Moreover, the amounts of these fertilizers were also differed in different locations due to the farmer accustomed fertilization. However, for the paired yield-invigorating and yield-debilitating orchards, the total amount of applied chemical fertilizer was similar but more organic fertilizer (usually two or three folds compared the amount of organic fertilizer applied to the yield-debilitating orchards) was applied into the yield-invigorating orchards according to the farm record. We have added the formation about the fertilization regimes as Table S2 in the supplemental material.

[17] L168 Provide references for all statistical software and packages used.

Response: Thanks for your suggestion. We have added it in the revised manuscript.

[18] L204-L208 When applying structural equation modelling it is good practice to provide a conceptual meta-model summarising underlying theoretical pathways, plus a table providing references to the hypothesised pathways. Please consider adding this information. Additionally, you should include a fit index, which is robust to sample size, such as the comparative fit index. Consider citing Grace (2006). **Response:** Thanks for your suggestion. We have added the fit index in the figure. The reference was also

cited in the revised manuscript. Moreover, a conceptual meta-model summarising underlying theoretical pathways was also added in the supplemental material.

**Results**

[19] L222-L223 The PCoA analysis shows clustering between sites. However, the PCoA does not clearly separate the treatments (Fig 2A). Revise.

Response: Thanks for your suggestion. We have revised the "treatment-based" as "location-based".

[20] L237-L241 Long sentence. Revise.

**Response:** Thanks for your suggestion. We have revised it as following: "A total of 2546 unique ASVs with 53,222 sequences were found in all yield-invigorating orchards while 2308 unique ASVs with 44,389 sequences were observed in all yield-invigorating orchards. Among these unique ASVs, almost 70% of ASVs were shared between yield-invigorating orchards and -debilitating orchards.".

[21] L244 Please check https://ncbiinsights.ncbi.nlm.nih.gov/2021/12/10/ncbi-taxonomy-prokaryote-phyla-added/ for new standards in taxonomic names.

Response: Thanks for your suggestion. We have revised it through our manuscript and figures.

[22] L258-L262 Move to M&M.

Response: Thanks for your suggestion. We have moved this sentence to Material and Method section.

[23] L272 What are the threshold values Zi and Pi?

**Response:** Thanks for your suggestion. The threshold values of *Zi* and *Pi* was 0.62 and 2.5, respectively. And we have added this information in the Material and Method section.

[24] L289-L291 Why is pH not included. Not explained so far. And probably not meaningful to exclude soil pH, due to it's important influence on microbial community composition.

**Response:** Thanks for your comment. As we all know, soil pH was s main driver for the assemblage of soil bacterial community in many cases. However, we used forward stepwise to build a module explaining the variation of soil microbial community. After selection, the modules include soil OM, TN, alkaline N, AP and AK, available calcium (Ca), copper (Cu) and manganese (Mn) explained the majority of the variation in bacterial community composition. Other soil properties were removed probably due

to the collinearity or not important roles in this study. As we discussed in the manuscript, the variation of soil pH was not very large in the agricultural soils. This maybe the reason why pH was not in the module.

[25] L291 RDA1 does only explain 17.4% of variation in bacterial community composition this does not support a strong influence of OM on community composition. Revise wording.

**Response:** Thanks for your comment. We have revised it as "As evidenced by the RDA vectors, OM within the module was identified as the top soil property that determines the composition of bacterial community.".

**discussion**

[26] L311 I don't agree that Fig.1C suggests a direct significant correlation between bacterial community and pear yield. Why do you assume it to be direct? Both could be correlated to SOM. The distinct bacterial community composition was not associated to yield but to treatment (as far as I understood). Yield differed across sites.

Response: Thanks for your comment. We have deleted the word of "directly".

[27] L316 Shared across six sites?

Response: Thanks for your comment. We have deleted the word of "Shared".

[28] L317 Well-organized not neutral à "more interactive"?

Response: Thanks for your suggestion. We have revised the it as your advice.

[29] L332 Delete "always".

Response: Thanks for your suggestion. We have revised it.

[30] L336-L338 Incomplete sentence.

Response: Thanks for your comment. We have revised it.

[31] L339 Define what you mean by soil "biological" fertility.

**Response:** Thanks for your comment. We have revised it. Here we want say that Planctomycetota may promote plant production through improving soil fertility, *i.e.*, involving nitrogen cycling. Therefore, we delete the word of "biological" to make it clearer.

[32] L345 See my major concern on the conclusion drawn on causality.

**Response:** Thanks for your comment. As we responded to the comments [2], we decide to delete this sentence.

[33] L354 What do you mean by "small world"?

**Response:** Thanks for your comment. We have added this information in this paragraph.

[34] L355 Revise conclusions.

**Response:** Thanks for your comment. We have revised it as "Our comparative network analysis indicated that microbial co-occurrence patterns in soils links to different pear production.".

[35] L361 Basal shifts in network architecture linked to fertilisation regime and or OM quality? **Response:** Thanks for your comment. We have no evidence about this question. However, we agree with you that the basal shifts in network architecture linked to fertilisation regime and/or OM quality. Anyway, this needs farther works.

[36] L365 Which study does "this" refer to?

Response: Thanks for your comment. We have revised it as "In our study".

[37] L370 How does the fertilisation regime in the cited study relate to the fertilisation regime in the presented study?

**Response:** Thanks for your comment. In the cited reference, the soil was fertilized with different treatments, including miner fertilizers, miner fertilizers plus organic fertilizer. The result was conclude based on the integrated result of these soils from different treatments. In our study, soils were fertilized with miner fertilizer plus organic fertilizer too. These two studies were roughly similar.

[38] L380-L374 Too speculative. Revise.

**Response:** Thanks for your comment. Considering soil pH was introduced in the Introduction section and the comments from another review, we have deleted this sentence.

[39] L378 Two points on a log-scale already are a huge difference in terms of soil pH. However, more important than the range across sites would be the difference between treatments – which looks small, indeed. However, best would be to include pH in the analysis and to show, that it really plays a minor role in determining bacterial community upon treatments.

**Response:** Thanks for your comment. As we all know, soil pH was s main driver for the assemblage of soil bacterial community in many cases. However, we used forward stepwise to build a module explaining the variation of soil microbial community. After selection, the modules include soil OM, TN, alkaline N, AP and AK, available calcium (Ca), copper (Cu) and manganese (Mn) explained the majority of the variation in bacterial community composition. Other soil properties were removed probably due to the collinearity or not important roles in this study. As we discussed in the manuscript, the variation of soil pH was not very large in the agricultural soils. This maybe the reason why pH was not in the module. Considering the comments from another review, we have deleted this sentence.

[40] L380 Not only a question of higher SOM content, but probably also associated to quality. Discuss.**Response:** Thanks for your comment.

**Response:** Thanks for your comment. We totally agreed with your comment. However, we did not measure the quality of soil organic matter in this study. Anyway, we have revised this sentence to discuss this point as following: "Despite the quality of soil organic matter was not evaluated in this study, the quality of soil organic matter was associated with the diversity of microbial community (Ding et al., 2015), which implies more attentions should be paid to illustrate the relationship between the quality of soil organic matter and microbial community in our future work.".

[41] L384 Fierer 2004 does not state that diversity increases with higher C content. He states that diversity depends on SOM quantity and quality.

Response: Thanks for your comment. We have revised it as your suggestion.

[42] L385 see comments on SEM.

**Response:** Thanks for your comment. We have responded to this comment in the response to comment [3].

[43] Fig 1 Separate A from B and C and make it an own figure. Does not belong to results.

Figure 1 C Why are there so many dots? If microbial composition distance was calculated for each replicate, it should be 48 dots. Additionally, yield data only represents orchard average per tree, thus it should be only 6 points (or lines, where the 48 dots of microbial composition distance align). Why are there more than 6 levels of yield distance? The relationship does not look linear. Consider choosing another colour for the dots.

**Response:** Thanks for your comment. We have revised the figure as new figure 1 and figure 2. The reason for many dots are mainly because the figure was plotted based on the baycurtis distance (a kind of beta diversity, the result was a matrix) not simply linear correlation.

[44] Fig 2 Shapes in A too small. Do not abbreviate yield-invigorating and yield-debilitating wherever the space allows to write the name (plus consider changing the treatment names).

Figure 2 A The PCoA analysis shows clustering between sites. However, the treatments does not clearly separate the treatments. Revise L222-L223.

Response: Thanks for your comment. We have revised it according to your suggestions.

[45] Figure 3 B Consider moving this to supplement. Does not add much to the main story.**Response:** Thanks for your comment. We have removed this part into the supplemental material.

[46] Fig 4 You show many results and provide a lot of valuable information. However, I am not sure how this Figure adds to the research question. Additionally, size too small. Hard to read.

**Response:** Thanks for your comment. Network analysis could provide very useful information about the potential interactions among the soil microbiome, which have been shown that play key roles in determining soil ecosystem functions. Therefore, we report this result in our manuscript. Following your suggestion, we have revised our manuscript to make it easier to read.

**[46] Fig 5 Important figure.**

Figure 5 A Implications are a bit unclear for Fig 5 A. Does this implicate if yield only explains 2% of bacterial community composition variation, only 2% of yield can be explained by the community composition?

Figure 5 B Soil pH should be included. See previous comment on that.

Figure 5 D Again, why are there so many points?

Response: Thanks for your comment. In this study, we firstly used the VPA analysis to show soil chemical properties play key roles in determining soil microbial communities. Next we used RDA analysis and stepwise to select key soil chemical properties driving soil microbial communities. Meanwhile, we used random forest analysis to explore the key soil chemical properties in predicting pear yield. Combining all these results together, soil organic matter was recognized as most important factor in driving soil microbial community and determining pear yield. For soil chemical properties, soil organic matter was only the factor selected to build the SEM mainly due to the following reasons: 1) only OM was significant differed between YI and YD soils, 2) OM was top parameters for predicting the orchard yield, 3) OM within the module was identified as the top important soil property that determines the composition of bacterial community as evidenced by the RDA vectors. Therefore, soil pH or another factors were not shown in the RDA result after VIF selection using vegan package. As for microbial community, the result of VPA showed soil chemical properties together may explain 36.8% variations of microbial community while location and yield only explain a small proportion. In our opinion, this probably location and yield were not direct factor in shaping the changes of soil microbial communities. They are mainly affected the composition of soil microbial communities first through changing soil properties. And our results are in line with to many previous VPA results that soil chemical properties explaining the most of the variations of microbial communities (Wu et al., 2021, 161: 108374; Yang et al., 2017, 215: 756-765; Zhao et al., 2013, 67: 443-453). Anyway, we have added this point in our Discussion section. The reason for many dots are mainly because the figure was plotted based on the baycurtis distance not simply linear correlation. And the regression was also revised in the manuscript.

[47] Fig 6 Structural equation model seems a good approach to address the question whether bacterial community composition improves pear yield. However, to address this question a direct path from SOM

to yield must be included in the model. Additionally, soil pH as a major driver of community composition, should be included as well, even if hypothesised to play a minor role in this study. The major lack of this SEM is that no information on R2 is provided. How much of the yield is explained by the SEM? All these points are needed to address the cited research question and to draw the conclusion drawn. Additionally, what does beta diversity PCoA mean? Unclear. The comparative fit index, which is not sensitive to sample size should be included. It remains unclear, how the model was developed and fitted.

**Response:** Thanks for your comment. The R2 is 0.739 for this SEM and we have added it in the revised manuscript. The raw data used for SEM construction was selected because these index may play key roles in supporting a high-yielding pear production based on the previous result in the present study. For soil chemical properties, soil organic matter may play the most important roles as different analysis pointed. As for microbial indices, the microbial abundance, key taxa and interactions were selected due to they were recognized as potential key components in determining pear yield. The detailed information about the parameters used for SEM construction was added in the revised manuscript. Following your suggestion, we also added a conceptual model in the supplemental material.

---

## Referee Report (RR1)

The authors of the manuscript soil-2021-95 "Soil microbial community triggered by organic matter inputs associates with a high-yielding pear production" carefully revised the manuscript and supplementary materials according to the reviewers' suggestions, particularly in regards to the concerns that data does not allow to draw the conclusion that distinct microbial communities directly increase pear-yields. There are still some minor concerns I recommend to consider.

Furthermore, after revisiting the structural equation model again, I recommend to undertake a major revision of the SEM or to delete it. In general, the M&M section to the SEM does not allow to reproduce and understand how the SEM was built and whether it clearly supports the research question or not. However, I am not convinced that the SEM was built in a way it allows to identify what drives the differences in pear yield between the treatments, which was what the study aimed for, as indicated in the introduction and L 341. This is due to the following reasons:

1) From the M&M it is not transparent whether the paired approach was accounted for (for example by including a random effect for site using the lavaan.survey package). If not, the model will try to explain overall yield, not the increase or difference between treatments.
2) Across site variances in SOM content and pear yield are larger than between treatment differences.
3) The SEM Figure 7 revealed that SOM has an effect on yield mediated through Beta. However, beta diversity by PCoA largely differed across sites and did not show differences between treatments (Figure 3A) as previously commented. Which support my assumption, that the model is fit to explain overall yield differences, not treatment related yield increases. (Also, I miss information in the M&M on whether beta diversity (PCoA) in the SEM refers to PCoA1 or PCoA2 or is a composite of both.)
4) Overall, the SEM explained > 70% of variance in yield. However, it might only explain differences across sites but does not allow to conclude on how differences in soil organic matter content affect yield of yield-debilitating and yield-invigorating orchards.

The SEM has to be revised or the results and discussion section adjusted to indicate that the SEM explains differences across sites. The model as it current state does not allow the conclusion in L 451 – 455: "Worth to mention, soil organic matter was not directly linked to the yield in the constructed model, indicating that soil organic matter maintain the high-yielding pear production probably via the indirect way. Therefore, we proposed that yield-invigorating soils harbour unique bacterial communities that may improve soil biological fertility, which could be driven by soil organic matter and manipulated by keystone species (Chloroflexota) through altering the bacterial interactions."

L 71          Rousk et al. 2010 did not investigate microcosms (in the cited study).

L 98 - 105    Thanks for clarifying the definition of "yield-debilitating" and "yield-invigorating"
              treatments. Please revisit this newly written part for proof-reading and grammar
              checking. I recommend adding the mean ± standard deviation increase in pear yield
              to estimate the order of magnitude. While revisiting your manuscript, I wondered

why you did not include average crop-yield orchards as a control. You might want to add a comment on that.

L 163    introduce abbreviation as follows:  Amplicon Sequence Variant (ASV)

L 231    This is not the reference I was referring to and citing the indicated study here is not meaningful. Sorry, I should have specified: Grace JB (2006) Structural Equation Modeling and Natural Systems. Cambridge University Press, Cambridge. Revise!

L 232    Add criterion for goodness-of-fit index.

L 243 - 246    Would be interesting to give more details on which chemical soil properties differed between YI and YD. I assume it is only organic matter content? Please, indicate statistical results which support that OM was higher in YI than in YD.

L 401 - 406    Thanks for adding a paragraph on what you mean by "small-world". This is helpful.

L 421    Revise! "…, was recognized as a key phylum in improving pear yield" → "…, was recognized as key phylum associated with higher pear yields".

Figure 7    Also see my comment on the SEM. I apologize I had overseen the direct path from SOM to yield, which was already included in your model in your initial submission. Providing an R2 for the whole model seems not meaningful. Instead, the R2 should be indicated for each endogenous variable in your model (i.e. beta diversity, microbial biomass, key taxa, network module hubs, network connectors and yield) indicating the variance explained by your model for each parameter.

Table S2    Thanks for providing a table on fertilisation regime. However, it is a bit confusing. Did YI and YD receive the same amounts of mineral fertilisers or do N,P,K values refer to a total (including N,P,K in organic fertilisers)? Please revise the Table so it becomes clear how much mineral fertiliser was added, and how much organic fertiliser. From the text I understood that YI orchards received less mineral fertiliser. Is that correct?

Figure S6    This is not the meta-model I suggested, however it helps to understand the underlying concept or your model.

---

## Author Response (AR2)

**Response to reviewer's comments**

The authors of the manuscript soil-2021-95 "Soil microbial community triggered by organic matter inputs associates with a high-yielding pear production" carefully revised the manuscript and supplementary materials according to the reviewers' suggestions, particularly in regards to the concerns that data does not allow to draw the conclusion that distinct microbial communities directly increase pear-yields. There are still some minor concerns I recommend to consider.

[1] Furthermore, after revisiting the structural equation model again, I recommend to undertake a major revision of the SEM or to delete it. In general, the M&M section to the SEM does not allow to reproduce and understand how the SEM was built and whether it clearly supports the research question or not. However, I am not convinced that the SEM was built in a way it allows to identify what drives the differences in pear yield between the treatments, which was what the study aimed for, as indicated in the introduction and L 341. This is due to the following reasons:

1) From the M&M it is not transparent whether the paired approach was accounted for (for example by including a random effect for site using the lavaan.survey package). If not, the model will try to explain overall yield, not the increase or difference between treatments.

2) Across site variances in SOM content and pear yield are larger than between treatment differences.

3) The SEM Figure 7 revealed that SOM has an effect on yield mediated through Beta. However, beta diversity by PCoA largely differed across sites and did not show differences between treatments (Figure 3A) as previously commented. Which support my assumption, that the model is fit to explain overall yield differences, not treatment related yield increases. (Also, I miss information in the M&M on whether beta diversity (PCoA) in the SEM refers to PCoA1 or PCoA2 or is a composite of both.)

4) Overall, the SEM explained > 70% of variance in yield. However, it might only explain differences across sites but does not allow to conclude on how differences in soil organic matter content affect yield of yield-debilitating and yield-invigorating orchards.

The SEM has to be revised or the results and discussion section adjusted to indicate that the SEM explains differences across sites. The model as it current state does not allow the conclusion in L 451 – 455: "Worth to mention, soil organic matter was not directly linked to the yield in the constructed model, indicating that soil organic matter maintain the high-yielding pear production probably via the indirect way. Therefore, we proposed that yield-invigorating soils harbour unique bacterial communities that may

improve soil biological fertility, which could be driven by soil organic matter and manipulated by keystone species (Chloroflexota) through altering the bacterial interactions."

**Response:** Thank you for your warm reminder. After fully discussion with all co-authors, we found that that SEM was not well described in the manuscript, and the SEM result does not help to much in explaining the potential mechanisms regarding to yield improvement. Therefore, we decide to delete this result and relevant material and discussion in the revised manuscript following your suggestion.

[2] L 71 Rousk et al. 2010 did not investigate microcosms (in the cited study).

**Response:** Thank you for your warm reminder. We have deleted the microcosms in the revised manuscript.

[3] L 98 - 105 Thanks for clarifying the definition of "yield-debilitating" and "yield-invigorating" treatments. Please revisit this newly written part for proof-reading and grammar checking. I recommend adding the mean ± standard deviation increase in pear yield to estimate the order of magnitude. While revisiting your manuscript, I wondered why you did not include average crop-yield orchards as a control. You might want to add a comment on that.

**Response:** Thank you for your warm reminder. We have revised the relevant sentences in the revised manuscript. We have added the standard deviation in the Table S1. In the present study, we aim to explore the key factors of soil chemical properties, and microorganism responding to high pear yield production. Indeed, the orchard with average yield production should be taken into consideration as another control. When designing this study, we thought by comparing the soils sampled from yield-debilitating and yield-invigorating orchards may be easier to achieve this goal at that time. Therefore, we only sampled these co-located yield-debilitating and yield-invigorating orchards with the same nutrients input. We sincerely appreciate your suggestion and we will take this point into consideration in our further works.

[4] L 163 introduce abbreviation as follows: Amplicon Sequence Variant (ASV)

**Response:** Thank you for your warm reminder. We have revised it following your suggestion.

[5] L 213 This is not the reference I was referring to and citing the indicated study here is not meaningful. Sorry, I should have specified: Grace JB (2006) Structural Equation Modeling and Natural Systems.

Cambridge University Press, Cambridge. Revise!

**Response:** Thank you for your warm reminder. We have revised it following your suggestion.

[6] L 232 Add criterion for goodness-of-fit index.

**Response:** Thank you for your warm reminder. We have revised it following your suggestion.

[7] L 243 - 246 Would be interesting to give more details on which chemical soil properties differed between YI and YD. I assume it is only organic matter content? Please, indicate statistical results which support that OM was higher in YI than in YD.

**Response:** Thank you for your warm reminder. We have revised it as following:

"Soil chemical properties differed among the locations and orchard yield types (Table S3). On average, yield-invigorating orchards showed obviously higher contents of OM, AP, Mg and Fe, and a lower content of Mn in comparison with those in yield-debilitating orchards. However, when taken all sites together, only a higher relative abundance of OM on average was observed in yield-invigorating orchards compared to that in yield-debilitating orchards based on the Wilcoxon test ($P < 0.05$)."

[8] L 401 - 406 Thanks for adding a paragraph on what you mean by "small-world". This is helpful.

**Response:** Thank you for your warm comment.

[9] L 421 Revise! "…, was recognized as a key phylum in improving pear yield" ◊ "…, was recognized as key phylum associated with higher pear yields".

**Response:** Thank you for your warm reminder. We have revised it as following:

[10] Figure 7 Also see my comment on the SEM. I apologize I had overseen the direct path from SOM to yield, which was already included in your model in your initial submission. Providing an R2 for the whole model seems not meaningful. Instead, the R2 should be indicated for each endogenous variable in your model (i.e. beta diversity, microbial biomass, key taxa, network module hubs, network connectors and yield) indicating the variance explained by your model for each parameter.

**Response:** Thank you for your warm reminder. We have deleted the SEM section in our manuscript as your suggestion. Please see the response to comment [1].

[11] Table S2 Thanks for providing a table on fertilisation regime. However, it is a bit confusing. Did YI and YD receive the same amounts of mineral fertilisers or do N,P,K values refer to a total (including N,P,K in organic fertilisers)? Please revise the Table so it becomes clear how much mineral fertiliser was added, and how much organic fertiliser. From the text I understood that YI orchards received less mineral fertiliser. Is that correct?

**Response:** Thank you for your warm reminder. For each site, the YI and YD orchards received the same amounts of nutrients. But they differed in organic and chemical fertilizer application. YI orchards usually received more organic fertilizer while YD orchards received more chemical fertilizer. We are so sorry about the unclear information, and we have added the detailed fertilizer amounts in the Table S2 in the supplemental material.

[12] Figure S6 This is not the meta-model I suggested, however it helps to understand the underlying concept or your model.

**Response:** Thank you for your warm reminder. We have deleted the SEM section in our manuscript as your suggestion. Please see the response to comment [1]. Therefore, this supplemental figure also was removed from the supplemental material.